# Epidemiological Profile among Greek CrossFit Practitioners

**DOI:** 10.3390/ijerph20032538

**Published:** 2023-01-31

**Authors:** Konstantinos Vassis, Athanasios Siouras, Nikolaos Kourkoulis, Ioannis A. Poulis, Georgios Meletiou, Anna-Maria Iliopoulou, Ioannis Misiris

**Affiliations:** 1Faculty of Physiotherapy, School of Health Sciences, University of Thessaly, 35132 Lamia, Greece; 2Department of Computer Science and Biomedical Informatics, School of Science, University of Thessaly, 35132 Lamia, Greece; 3“Τhe Warehouse Gym”, 62100 Serres, Greece; 4“Meletiou_PhysioLab” Advanced Physiotherapy Center, 32200 Thiva, Greece; 5“Physioclock” Advanced Physiotherapy Center, 41222 Larissa, Greece

**Keywords:** CrossFit training, epidemiological profile, injury, risk factors

## Abstract

CrossFit (CF) is a popular and rapidly expanding training program in Greece and worldwide. However, there is a lack of scientific evidence on the risk of musculoskeletal injuries related to CF in the Greek population. A self-administered survey of 1224 Greek CF practitioners aged 18 to 59 was conducted and analyzed using the Statistical Package for Social Sciences (SPSS) software. The highest percentage of the participants (34%) practiced 5 days per week for 60 min (42.2%) and had 2 days per week of rest (41.7%). A total of 273 individuals (23%) participated in CF competitions and 948 (77%) did not. The results showed that the most common injuries were muscle injuries (51.3%), followed by tendinopathies (49.6%) and joint injuries (26.6%). The shoulders (56.6%; n = 303), knees (31.8%; n = 170), and lumbar spine (33.1%; n = 177) were the most commonly injured locations. The logistic regression model showed that participation in competitions (*p* = 0.001), rest per week (*p* = 0.01), duration of training per session (*p* = 0.001), and frequency of training per week (*p* = 0.03) were statistically significant factors for injury. Training level was not a statistically significant factor for injury (*p* = 0.43). As CF continues to gain popularity on a global scale and the number of athletes gradually increases, it is important to monitor the safety of practitioners. Clinicians should consider participation in competitions, rest, training duration, and frequency in order to make CF safer.

## 1. Introduction

Pursuing health and sports avocations has become increasingly popular, and exercise is considered a critical component of maintaining long-term health. People have increasingly turned to various activities to gain the benefits that exercise can offer [1]. CrossFit (CF) is one of the most popular and rapidly growing exercise regimens in Greece as it is worldwide [2]. It is a high-intensity conditioning and training program [3,4] that was invented by Greg Glassman in the year 2000 as a trademark of CF^®^ [5] and became a sport in 2007 when the workout program became competitive with the introduction of the CF Games [5].

CF has gained popularity and interest among physically active populations [4,6,7,8] for its focus on successive ballistic motions that build strength and endurance [4,9]. Today, there are over 15,000 certified and registered CF centers (called “boxes”) all over the world [10], out of which 15 are located in Greece [11]. Nonetheless, there are also unregistered gyms that offer CF but do not meet the instructor requirements and do not pay the license to use the name “CrossFit^®^” [12]. The number of unregistered gyms and participants is unknown in Greece.

CF incorporates a physical exercise philosophy, a competitive fitness sport, a peer-supporting atmosphere, and, for some people, a lifestyle [2]. It is based on a set of complex functional exercises, including three basic types: (1) cyclical (running, rowing and rope jumping), (2) weightlifting (Olympic weightlifting and basic weightlifting), and (3) ballistic movements (bars, push-ups, etc.) [3,13,14]. These exercises are performed at a high intensity, quickly and repetitively, with little or no recovery time between sets [4,7,8,14], which is associated with rapid muscular fatigue and may lead to a loss of concentration and skill, increasing the risk of injury [7,8,15].

Recently, due to the rapid increase in compliance with CF practice, studies have been conducted related to injury epidemiology in CF practitioners in many countries, including Italy [16], Brazil [6,17,18,19], Dutch [20] Portugal [3,6], France [21], USA [2,4,7,22,23], South Africa [24], Costa Rica [25], etc. It is necessary to assemble more data about musculoskeletal injuries from this type of exercise in order to understand and record them, recognize and reduce the risk factors for injury, increase the safety level, and better serve the population that participates in CF. All these will provide us with information to establish preventive strategies and further study CF epidemiology and injuries [4].

To the best of our knowledge, since there have been no such epidemiological studies to date, this is the first epidemiological study on CF practitioners in Greece. Thus, this study aimed to 1. investigate the epidemiological profile, sports history, training routine, habits, and presence of injuries among CF participants in Greece; 2. identify the most common musculoskeletal injuries endured during CF; and 3. determine the main risk factors for musculoskeletal injuries in Greek CF practitioners.

## 2. Materials and Methods

### 2.1. Study Design

This study is an observational, descriptive, and epidemiological study in order to collect data on musculoskeletal injuries in CF practitioners in Greece. The study was approved by the Ethical Committee of the Faculty of Physiotherapy, School of Health Sciences, University of Thessaly (n.1575ΣΕ2/13-4-2020). From April to July 2020, data were collected using an electronic version of a survey tool and a Google-based form in the Greek language (see Appendix A). All participants were fully informed about the assurance of anonymity, the purpose of the study, how their data will be used, and if there are any associated risks.

### 2.2. Population

The study sample consisted of CF practitioners of both sexes who worked out in Greece regardless of whether they participated in competitions according to the inclusion and exclusion criteria set by the authors. The questionnaire was distributed to CF boxes all over Greece provided that boxes had at least Level 1 certified trainers. In order to find the registered and affiliated CF boxes in Greece, we visited the website “maps.crossfit.com”. Coaches were encouraged to send our questionnaire to their athletes. A link to our survey was also posted in the active Facebook group “CrossFit Greece.” and to other similar pages on social media.

The study was carried out using a convenience sample and the snowball sampling method, including athletes who trained in CF-affiliated boxes in Greece. A convenience sample is a very common and practical approach used for sampling in similar studies. This method selects whichever volunteer is available and meets the inclusion criteria [26]. In the snowball sampling method, individuals with predefined characteristics initially selected for the survey provide contact information and recommend to the researcher other volunteers who meet the selection criteria based on the contacts that these individuals have. It is a method used to quickly increase the sample size of research, which is why the concept of snowball constantly grows as it goes down [27].

The inclusion criteria were determined to include CF practitioners who (i) practice in Greek boxes with at least one Level 1 certified trainer; (ii) are aged 18 or older; and (iii) have attended at least one one-week training session and were present during the data collection days. Participants were excluded if they (i) trained independently outside of a CF gym or outside Greece; (ii) trained in noncertified fitness centers or on their own; (iii) were performing other types of functional training; (iv) were younger than 18 years; (v) were affected by chronic osteoarticular diseases; and (vi) provided incomplete questionnaires.

### 2.3. Sample Calculation

The following simple formula (1) was used for calculating the adequate sample size in our prevalence study:(1)$$n\geq \frac{1.96^{2}}{\delta^{2}}p\left(1-p\right)$$
where *n* denotes the sample size, *p* the estimated prevalence, and *δ* the margin of error. The minimum sample size required to conduct the study was 1068 people, taking into account that there exist approximately 35,000 CF athletes all over Greece and assuming that 50% are injured, with 95% CI and 3% accuracy of injury prevalence. Since the existing number of CF participants in Greece remains unknown, the authors of this study followed the calculation method of Sprey et al. [6].

### 2.4. Measurement Instrument

Since there are no validated questionnaires on this topic, an electronic, anonymous, and self-administered questionnaire served as the measurement instrument. The questionnaire was based on previous relevant studies [3,6,16,20] and was further modified by four experts with different backgrounds. Participation was on a voluntary basis, and participants did not receive money to respond to the questionnaire. Coaches, owners, and participants were further encouraged to send the survey to members of other gyms according to the inclusion criteria.

The questionnaire consisted of three parts (Appendix A). The first part included items concerning demographics—general information (gender, age, height, weight, and training location). The second part had specific questions about CF (physical effort that work involves, CF experience, training sessions per week and duration, participation in competitions, days a week of rest, regular performance of other sports, sports activity level prior to CF, reasons for starting to practice CF, supervision by a coach, professional monitoring, recovery, nutrition, etc.). The last, third part focused on the occurrence of injuries while practicing CF, injury location, and type of injury over the whole CrossFit practice (lifetime prevalence).

For better standardization, the definition of “sports injury” is defined as in studies by Weisenthal et al. [4] and Sprey et al. [6]. It was considered the only situation that had resulted in one or more of the following: 1. any physical complaint severe enough to necessitate the individual seeking medical attention to diagnose or treat the injury; 2. modification of normal training activities in terms of duration, intensity, or mode for more than two weeks; and 3. more than one week of absence from CF training and other outside physical activity routines.

### 2.5. Statistical Analysis

Statistical analysis was performed with Statistical Package for Social Sciences (SPSS), version 25.0 (IBM Corp.: Armonk, NY, USA), including t-tests for independent variables and one-way ANOVA. Quantitative variables are described as means with standard deviations; qualitative variables are described as percentages reporting 95% confidence intervals. Before proceeding to multivariate analysis, univariate analysis was performed on all variables. A logistic regression model was created in order to examine the risk factors for injuries in CF. Diagnosis of the logistic regression model was made, and multicollinearity and interactions were evaluated. The chi-square test was used to test the significance of the single-factor analysis. All *p*-values were 2-tailed with confidence intervals (95% Cls). The level of statistical significance was set at 0.05.

## 3. Results

A total of 1224 CrossFit practitioners, 443 females (36%) and 781 males (64%), comprised the study’s sample (value above the estimated minimum sample size), aged between 18 and 59 years (30.76 ± 7.58 years), with the following characteristics: mean height 175.40 ± 8.78; weight 75.89 ± 13.24 kg; and BMI 24.5 ± 2.72 (Table 1). The majority of participants (41%) reported that their work was essentially sedentary; 33% had work that was distinguished by many hours of standing and walking without physical exertion; and 23.4% had work that required a lot of standing. Only 2.6% of the workers did heavy lifting (Table 1).

Regarding sports activities, 1069 practitioners (87%) reported that they participated in sports before starting CF. The main sports activity that was described was weight training (430 athletes, 35.1%), followed by soccer (145 athletes, 11.8%), basketball (80 athletes, 6.5%), running (64 athletes, 5.2%), and swimming (57 athletes, 4.7%). When asked about the level at which they practiced these sports activities, 48.9% (598) reported that they practiced at a medium level, 33.5% (410) at a high level, and only 17.6% (216) trained at a low level (Table 2).

The athletes’ characteristics include 594 (48.5%) who had practiced CF for more than 2 years; 361 (29.5%) for more than 1 year and up to 2 years; 143 (11.7%) for a period between 6 months and 1 year; and 126 (10.3%) for up to 6 months (Table 3). Most of them (68%) started CF for fitness reasons, while 26% did so out of curiosity due to the popularity of the sport, and 22% did so on a professional’s recommendation. The majority of the participants trained more than four times per week (4,27 ± 1.14 days per week) and each training session was more than 1 h (92%) in duration (Table 3).

The most significant characteristics were compared between the two groups of 1224 CF athletes (534 injured and 690 healthy). Their demographics (sex, age, height, and BMI), CF training history, training time, weekly participation, competition participation, rest day, level of athletic activity, medical history, coach monitoring, health professional specialist, passive recovery, nutrition program, dietary supplements, and types of dietary supplements were compared between these two groups. Table 4 summarizes the outcomes of these comparisons along with the number for each group, the mean, and the standard deviation.

The most injured body regions were the shoulder (n = 305, 31.4%), lower back (n = 178, 18.3%), and knee (n = 169, 17.4%). On the other hand, the foot (n = 24, 2.5%), fingers (n = 10, 1%), and head (n = 1, 0.1%) were the less affected body regions (Figure 1).

When asked about the type of injury they had, the 256 athletes reported that they (49.6%) had a tendon injury, 231 (43.3%) had a muscle injury, 142 (26.6%) had a joint injury, and 73 (13.7%) had neuropathic pain, while 34 (6.4%) did not know what type of injury they had. A total of 310 (58.1%) of them had an injury to the same body region in the past. Furthermore, 419 (78.5%) were treated for their injury by a health care professional, doctor (49.6%), or physiotherapist (52.4%), and 385 (72.1%) of them followed rest as treatment, 298 (55.8%) physiotherapy, 147 (27.5%) medication, 63 (11.8%) injections, and 22 (4.1%) surgery (Table 5).

Figure 2 presents descriptives of units per risk factor. Statistically significant injury factors calculated by a logistic regression model were participation in competitions (*p* = 0.001); rest per week (*p* = 0.01); duration of training per session (*p* = 0.001); and frequency of training per week (*p* = 0.03). Training level (*p* = 0.43) was not a statistically significant injury factor (Table 6).

## 4. Discussion

CF is a demanding power-training program based on a variety of exercises with high-intensity, repetitive, and multijoint movements, and the possible risks of injuries that may occur from CF participation have raised concerns, especially when they are performed inappropriately [1,2]. To the best of our knowledge, this is the first study in the Greek CF population and one of the largest samples worldwide to examine CF injuries.

Greek CF practitioners, in general, are young individuals (30.76 ± 7.58 years old); most of them are physically active, with prior sports experience (87%) at medium (48.9%) and high (33.5%) levels of sports participation; and most of them started CF for fitness reasons (68%). They usually prefer CF over other monotonous conventional training methods [6].

### 4.1. Musculoskeletal Injury Analysis

To date, many international epidemiological studies have been conducted, with the prevalence of injury being controversial and ranging from a rate of 12.8% [8] to a rate of 73.5% [28]. In our study, 44% of our sample had at least one injury while practicing CF. Weisenthal et al. [4], in their survey of 386 CF practitioners in New York, showed 19.4% of injured athletes over a 6-month period; Montalvo et al. [7], in 191 CF athletes in South Florida, verified that 26.2% had an injury in the previous 6 months; and Feito et al. [22] revealed 30.5% of injuries in a 12-month period in 3049 CF participants. Similar to our findings, Mehrab et al. [20] demonstrated a value of 56.1%. On the other hand, Hak et al. [28] showed that 73.5% of CrossFit athletes sustained an injury during CF training.

The moderate prevalence obtained could be explained by the fact that a nonrandomized data collection method was used, and this may affect the results obtained. As a result, the questionnaire may have been filled out by more CF practitioners who had an injury in the previous period and thus might have felt more motivated to fill it out. Secondly, we did not set time restriction limits in the wording of the question about injuries—”did you have any injuries during CF? “—in order to quantify the number of cases of injury that occurred. In their studies, Weisenthal et al. [4], Feito et al. [22], and Montalvo et al. [7] only looked at survey respondents who reported having an injury within the previous six and twelve months, rather than the entire sample, when looking at injuries, while Hak et al. [28] did not restrict injury incidence to a specific time period. In our situation, the full sample was assessed, so it is possible that we cannot compare our findings about injury prevalence with those of other studies.

Greek practitioners had 106 injuries over the previous 12 months of CF participation. Minghelli et al. [3], in their study, showed 108 injuries, Weisenthal et al. [4] observed 84 injuries in a period of 6 months, and Montalvo et al. [7] verified a total of 62 injuries in CF participation over the same time period. Hak et al. [28] reported a total of 186 injuries during the entire CF practice. To make our findings comparable to those of other studies, we calculated the number of injuries based on those sustained in the previous year. Ιn order to perform this calculation, we selected participants with 0–1 years of experience in CF at the time they completed the questionnaire. Nevertheless, it appears from all the studies that the numbers are comparable to or lower than other sports such as Olympic weightlifting, distance running, military conditioning, track and field, rugby, or gymnastics [4,28,29]. Therefore, there are no data to support the claim that CF causes injuries in athletes more frequently than other comparable exercise regimens, suggesting that CF is a safe and useful form of exercise.

#### 4.1.1. Type and Body Location of Injuries

Concerning the type of the injury, the present study showed that the most common types were muscle injuries (51.3%) and tendinopathies (49.6%) followed by joint injuries (26.6%). Minghelli et al. [3] showed that joint injuries (cartilage, meniscus, ligament injury/sprain, and luxation) (30.8%) and muscle injuries (strain and contusion) (23.1%) were the most prevalent types of injuries. On the other hand, Szeles et al. [19] reported muscular injuries (45.34%), joint pain (24.7%), and tendinopathies (12.96%); Da Costa et al. [17] reported muscle strains (41%), overload injuries (26.2%), and contusions (17.3%); and Tafuri et al. [16] reported tendon injuries (16.7%), bruises (6.6%), and muscle contractures (30.6%). In the literature, some studies classified the type of injury in a variety of ways, making it difficult to compare with our findings. Weisenthal et al. [4] verified that the most common types of injuries were general inflammation and pain (30.8%), other (27.2%), and sprain/strain (17.2%), while rupture (3.7%) and dislocation (2.5%) were relatively infrequent. However, it seems that CF causes muscular injuries at a higher percentage, which may be due to its high-intensity nature and can range from simple muscle contractures to muscular lesions and rhabdomyolysis.

With regard to injury location, our results indicated that the shoulder (31.4%), lower back (18.3%), and knee (17.4%) were the most commonly injured body regions. Many epidemiological studies [2,3,4,7,8,17,19,20,22,28,30,31,32] have described injuries among CF athletes considering the most affected anatomical body parts. Our results were similar to the findings from Hak et al. [28] (shoulders (almost 45%) and spine (almost 35%)), Weisenthal et al. [4] (shoulder (25%), the lower back (14.3%), and the knee (13.1%)), Minghelli et al. [3] (shoulder (35.9%), the lumbar spine (17.9%), and the knee (11.5%)), Feito et al. [33] (shoulders (39%), back (36%), and knees (15%)), Mehrab et al. [20] (shoulder (28.7%), lower back (15.8%), and the knee (8.3%)), Szeles et al. [19] (shoulder (19%), lumbar spine (15%), and the knee (11.7%)), and Da Costa et al. [17] (shoulder (30.8%), low lumbar spine (30.1%), and leg (19.2%)). Similar results were also observed in Montalvo et al. ’s [7] study, where the highest percentage of injuries occurred in the shoulder (22.6%), followed by the knee (16.1%) and the lower back (12.9%). Moran et al. [8] showed that the most commonly affected body region was the lower back (33%), followed by the knee (20%), while the shoulder region had a lower incidence of injury (6.7%). Everhart et al. [30] reported knee (26.4%), shoulder (21%), spine (20.4%), and hip and groin injuries (8.5%); Larsen et al. [31] reported injuries of the lower back (25%), knee (21.4%), elbow/hand (17.9%), other anatomical locations (17.9%), shoulder (7.1%), neck (3.6%), hip (3.6%), and ankle (3.6%); Hopkins et al. [32] reported injuries of the spine (20.9%), shoulder (18.3%), and knee (15.5%); and Alekseyev et al. [2] reported back injury (32.2%) and shoulder injury (20.7%).

Our results and those of similar epidemiological studies reveal that the shoulder, lumbar spine, and knee joints appear to be commonly injured in CF practice. This is a possibility because these are the main body regions used in powerlifting, Olympic lifting, and gymnastic movements, which are the basic types of CF exercises [4]. In this sense, CF athletes can be considered powerlifters because CF includes exercises similar to weightlifting sports such as Olympic weightlifting [33]. For the sake of truth, Keogh and Winwood [33] showed in their study that Olympic weightlifters most frequently injured the knee, lower back, and shoulder; powerlifters most frequently injured the shoulder, lower back, and knee; and strongmen most frequently injured the lower back and shoulder.

Weisenthal et al. [4] observed that powerlifting movements resulted in more injuries than Olympic weightlifting movements (23% vs. 17%). Compared to powerlifters and CF athletes, Olympic weightlifters are more used to lifting weights overhead. As a result, they might be more skilled, stronger, and more flexible than other lifting athletes. According to Keogh et al. [34], elite Olympic weightlifters were less likely to sustain injuries than nonelite Olympic weightlifters. This finding suggests that increased skill, strength, and flexibility are associated with decreased injury incidence. All of these data imply that CF athletes should increase their skill, strength, and flexibility in overhead gymnastics and Olympic lifting sports if they want to lower their risk of shoulder injuries. Consequently, prevention programs should concentrate on the shoulder, lower back, and knee joints, and CF coaches should pay close attention to exercise execution.

#### 4.1.2. Treatment and Nutrition in CF Participants

Most of our study’s CF participants (78.5%) were treated by a health care professional or rehabilitation specialist. Specifically, 52.4% visited a physiotherapist and 49.6% a doctor; 72.1% followed rest as treatment followed by 55.8% physiotherapy and 27.5% medication. In the study by Minghelli et al. [3], the CF practitioners (71.8%) underwent some type of treatment; 46.4% of them did physiotherapy; 23.2% rested or took medication; 14.3% resorted to nonconventional therapies; 12.5% were immobilized for a period of time; 1.8% underwent surgery; and 1.8% performed another type of treatment. Everhart et al. [30], in their study, noted that 76% of injured subjects received physical therapy, 37.5% underwent advanced imaging such as magnetic resonance imaging, 15.8% received injections, and 15.8% underwent surgery. Other studies showed lower percentages of those who received treatment [2,6,7,28]. Alekseyev et al. [2] reported that 67.1% sought a health care professional to diagnose or treat the injury (received treatment for their injury). Similar were the results of Sprey et al. [6] who reported that 42% reported visiting a health care professional to diagnose or treat their injury, and Montalvo et al. [7] reported this figure to be 41.9%. Hak et al. [28] stated that 7% of injuries required surgical intervention, while in our study, only one individual was submitted to surgery.

The importance of sufficient nutrition for athletic performance is well documented [35]. In our study, 250 (20.8%) of the total CF athletes followed a nutrition program and 638 (52.1%) received dietary supplements. The types of supplements were vitamins (350 (28.6%)), followed by protein (340, 27.8%) and amino acids (250 (21.2%)), while some of them received other supplements (73 (6%)). However, only 139 (11.4 %) visited a nutritionist.

CF provides certification classes for four levels of trainers [36]. In CF-Level 3 and Level 4, trainers receive nutrition education with an emphasis on the Paleolithic diet [37]. In comparison, Level 1 and 2 CrossFit trainers are not required to have any nutrition education [38]. To the best of our knowledge, there are few trainers that are CF-Level 3 certified in Greece. Due to the increasing popularity of CF training, along with the vital role of nutrition in athletic performance, evaluating CF trainers’ nutrition knowledge is necessary. Maxwell et al. [38], in their online survey of 8875 CF trainers, studied 1. nutrition perceptions; 2. basic sports nutrition knowledge; 3. nutrition resources used; and 4. types of dietary advice given by CF trainers to their athletes. The results of this study pointed out that, despite the fact that CF trainers are aware of the importance of nutrition for athletic performance, their nutrition knowledge is not optimal. Additionally, they suggested that the trainers would benefit if nutrition education was included as part of the Level 1 and Level 2 CrossFit certification processes in order to promote their nutrition knowledge and, eventually, participants’ athletic performance.

Although there is a lack of scientific evidence, CF practitioners often adopt several strategies such as supplementation of medium-chain triacylglycerol (MCT), isolated amino acids, or buffering supplements (b-alanine and SB—sodium bicarbonate), as well as restrictive diets (Paleolithic or KD), believing that it will increase their exercise performance [39]. It should be underlined that the aforementioned nutrition strategies are controversial because they do not meet the recommendations of esteemed associations such as the International Society of Sports Nutrition (ISSN) [40]. It is suggested that more research is needed in order to obtain unambiguous information on nutrition in CF. It seems also justified to educate athletes and coaches about nutritional habits and individual energy and nutrient requirements [41].

### 4.2. Risk Factors

Concerning risk factors, in the literature, the risk factors for injury that have been proposed are the duration of participation in CF [2,6,7,17,19,20,22,25,42,43], the existence of previous injuries [8,17,19,44], the weekly training frequency [2,3,7,45], participation in competitions [2,3,7,17,25,42], and male sex [4,8,46]. Other factors such as advanced age [30], stretching before CF practice [2], alternating different training loads [19], not visiting a physical therapist on a regular basis [45], and coach supervision [4] have also been mentioned as associated risk elements. In our study, univariate analysis was performed, and several variables (time, weekly participation, participation in competitions, rest days, level of athletic activity, nutrition program, training level, and coach monitoring) were statistically significant. From all the experiments we conducted, we kept participation in competitions (*p* = 0.00), rest per week (*p* = 0.01), duration of training per session (*p* = 0.00), and frequency of training per week (*p* = 0.02), which were statistically significant risk factors for injury. Training level (*p* = 0.43) was not a statistically significant injury factor.

#### 4.2.1. Participation in CF Competitions

In addition to the risk of injury during workouts, in competitions, athletes face higher pressure to perform optimally in order to obtain the best results, which can put a significant strain on their physical capabilities [47]. As the culmination of an athlete’s long-term training is often competing in events [48], sports injury epidemiology generally agrees that there is a higher risk of injury during competitions than during training [48,49,50,51]. The parameter of whether a person participates in CF competitions appears to be an injury risk factor in the literature; however, determining whether competitors have a higher probability of being injured than those who do not compete, or the other way around, is still in progress.

In our study, of the individuals who reported at least one injury in CF (43.6%), 345 (28.2%) did not participate in competitions and 189 (15.4%) did. Regression analysis revealed that participation in competitions was a risk factor for injury (*p* = 0.001) and that athletes who do not participate in competitions are 2.9 times more likely to be injured than those who do. Our results are in agreement with the study of Minghelli et al. [3] and in contrast with the results of Montalvo et al. [7], Da Costa et al. [17], and Escalante et al. [25]. Specifically, Minghelli et al. [3] reported that athletes who did not participate in competitions had a 2.64 times higher probability of being injured than those who participated in competitions. In contrast, Montalvo et al. [7] and Sprey et al. [6] showed that competitors tend to suffer more injuries, with an injury incidence of 40% and 38.9% compared to 19.05% and 27.5% for noncompetitors [7], respectively [6]. Nevertheless, Sprey et al. [6] revealed that participation in CF competitions was an injury risk factor (*p* = 0.006) but not when analyzed as an isolated injury risk factor (*p* = 0.917). Alekseyev et al. [2] reported that advanced-level athletes (defined as capable of competing in CF games) were 2.63 times (*p* < 0.0001) more likely to be injured than intermediate-level athletes (defined as group-training multiple times per week but not at a competitive level) (n = 537) and 7.27 times (*p* < 0.001) more likely to be injured than beginner-level participants (defined as recently initiating solo training but not at a competitive level) (n = 240). Da Costa et al. [17] also found that the probability of injuries among competitive-level athletes was approximately five times higher than among beginners, and Escalante et al. [25] reported a significant relationship between getting injured and participating in CF competitions (*p* = 0.02).

Noncompetitors, as observed in this study, are participants who report a significantly lower number of training hours (frequency and duration) and less practice time than competitors (individuals who did not participate in competitions: 4.02 ± 1.14 times per week, 3.51 ± 1.39 h per session; those who competed: 5.12 ± 1.13 times per week, 5.08 ± 1.40 h). As a result, one explanation for our findings is that practitioners who train less have less technical proficiency to perform exercises correctly due to lower fitness levels and proper execution technique, making them more prone to sustaining injuries [3]. Furthermore, practitioners who compete are likely to have better coaching supervision and more training time, resulting in more correct technical execution and benefiting from a more individualized training prescription (exercises and load) [3].

#### 4.2.2. Rest Days per Week

Various epidemiological studies [4,6,23,25,52] in CF asked their participants about the number of rest days per week. Weekly rest days were a significant risk factor in our study, as evidenced by a significant relationship with injury incidence (*p* = 0.01). The likelihood of injury is 2.9 times higher for those who do not compete than for those who do in our CF population of athletes. This is in contrast to similar studies [6,23,25] that showed no significant relation between injury incidence and rest days per week.

#### 4.2.3. CF Training Duration

There is conflicting evidence regarding whether the duration of CF participation is a risk factor for injury. In our study, athletes who train up to 60 min per workout (including warm-up) are 0.8 times more likely to be injured than those who train >60 min. In CF athletes, Alekseyev et al. [2] discovered a positive relationship between injury frequency and weekly training duration. This supports the research by Montalvo et al. [7], which found an association between a longer period of CF engagement and a higher injury incidence. However, Soares [52], Sprey et al. [6], and Weisenthal et al. [4] found no correlation between injury incidence and the length of training sessions. More research is needed to determine whether the length of CF training has an impact on CF injuries.

#### 4.2.4. Weekly CF Training Frequency

Another significant risk factor for injury in CF is training frequency. The American College of Sports Medicine (ACSM) recommends that at least three training sessions per week be included in a training program in order to achieve physiological training effects such as improved cardiovascular conditioning and increased muscle size [53]. In our study, the risk of injury was 1.2 times higher for people who trained three or more times per week than for those who trained no more than twice. This comes in contrast with two previous studies [3,22]. Minghelli et al. [3] discovered that participants in CF who trained twice or less per week had a 3.24 times higher probability of injury than those who exercised more. Feito et al. [22] noted that practitioners who trained fewer times a week were more likely to sustain an injury than those who trained more frequently. Further research is needed to better understand the complex interplay between training frequency and injury risk in this population.

#### 4.2.5. Prior to CF Training Level

The training level before starting CF was not a statistically significant risk factor for injury in our study. The relationship between training level before CF and injuries has not been studied in the literature. Despite this, it could be argued that there is an indirect relationship because participation in a sporting activity may help the musculoskeletal system better manage loads. Of course, CF is not the same as sports such as soccer, basketball, and other sports in which our participants mentioned participating. Nevertheless, future studies should study whether sports-specific training may influence future CF injuries [54].

### 4.3. Limitations

Ιt is important to mention any potential limitations of the study as well as any biases that might have existed. Firstly, in Greece, there are many boxes that are not official CF affiliates. Official CF boxes have a set exercise program with specific scalability for different levels, whereas nonaffiliated boxes have little control over the training they provide. For this reason, we chose to give the questionnaire to at least Level 1 certified trainers in order to have a minimum level of coaching experience with standardized equipment and training and to ensure that the sample is as homogeneous as possible. However, because the survey was announced through social media, it cannot be completely checked that the questionnaire was not answered by some who did not meet the above criteria. As mentioned above, practitioners who had had an injury in the previous period might have felt more motivated to answer the questionnaire, so the high injury prevalence obtained might have influenced the results. As for the design of the studies, to date, there are many epidemiological studies that have many methodological differences (in the data collection period and method, in sampling method and size, etc.).

We acknowledge the limitations of our survey tool, even though we think our results offer valuable information regarding the epidemiological profile of Greek CF participants. Despite there being studies using similar questionnaires, there is no validated questionnaire with specific questions, so comparing studies and their results often becomes difficult. Additionally, a lack of randomization in our sample could have influenced our results.

It is challenging to compare the prevalence of injuries across research since different studies’ definitions of an injury vary. For example, while Weisenthal et al. [4] consider three specific criteria for what an injury consists of, Moran et al. [8] use a different definition. This fact can significantly influence the number of injured participants reported in each study and calls for the need for a universal definition of injury. Moreover, it might be difficult to determine whether an injury caused by CF is new or merely an aggravation of one that existed prior to beginning CF. Delayed muscular pain, which is very prevalent during high-intensity activities, may be mistaken for an injury. We do not know if all reported injuries were assessed by health professionals or if they were self-made diagnoses. Therefore, there may be disagreement regarding the accuracy of the athletes’ classification.

As far as the risk factors are concerned, in the majority of the studies (as in this study), typical statistical techniques were used to analyze the data, including multiple regression, Pearson correlation coefficients, and general linear models with partial correlation coefficients [55]. Regression analysis does not automatically “learn” from complicated data relationships because it is static and not predictive, especially when more data inputs are introduced [55]. Nevertheless, using this data set, Moustakidis et al. [55] made the first foray into the development of models capable of predicting CF injuries using cutting-edge machine learning (ML) algorithms, taking into account the massive proliferation of data in sports and addressing the growing needs for reducing the health, performance, and financial consequences of injuries in athletes. ML refers to the process by which a computer system utilizes data to train itself to make better decisions with significant potential for use in sports medicine research.

## 5. Conclusions

As the number of athletes gradually increases, the absolute number of injuries will consequently increase. Sports scientists should be familiar with CF and monitor the safety of practitioners to prevent injuries. Clinicians should consider participation in competitions, rest, training duration, and frequency to make CF safer for participants. A better understanding and identification of common injuries and potential risk factors of CF is needed to develop appropriate preventive strategies and reduce injury rates.

Further studies are needed to expand our knowledge of CF-related injuries and prevention and provide more detailed information on movements and exercises that commonly cause injuries. It could be good to explore means of reducing injury prevalence, especially in specific groups such as competitors and noncompetitors or beginner and intermediate-level athletes. Evidence-based recommendations can be made and included in CF routines, such as emphasizing proper technique, educating athletes about the risks of overexertion, or increasing the number of trainers or support staff at CF boxes.

## Figures and Tables

**Figure 1 ijerph-20-02538-f001:**
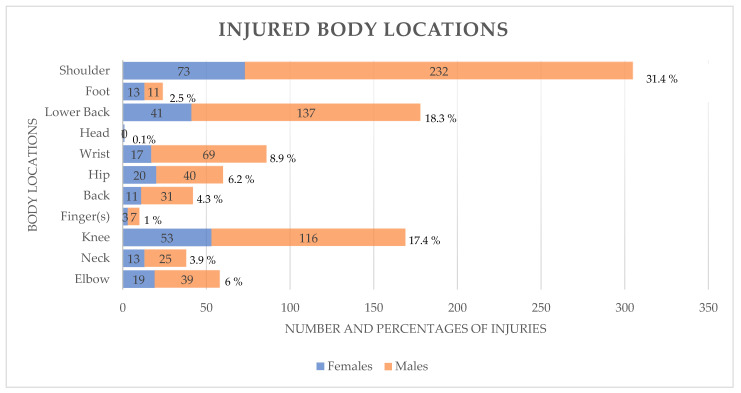
Injured Body Locations Graph.

**Figure 2 ijerph-20-02538-f002:**
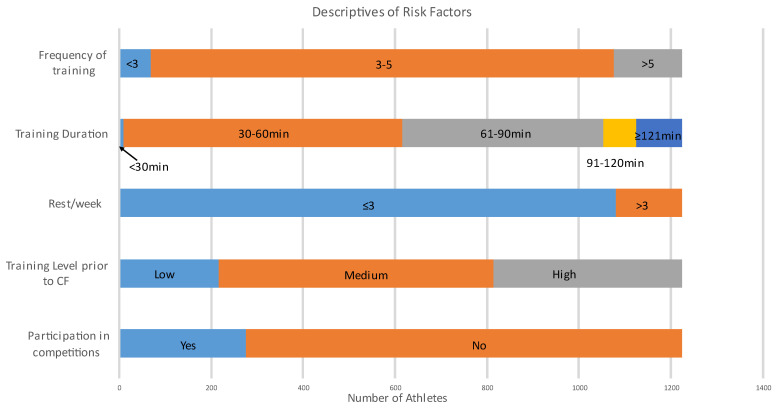
Descriptives of Risk Factors.

**Table 1 ijerph-20-02538-t001:** Demographic Data and Profile.

Sex	n (%)
Female	443 (36)
Male	781 (64)
**Age group**	
<18	0 (0)
18–29	598 (49)
30–39	448 (37)
40–49	162 (13)
≥50	16 (1)
Mean ± SD	30.76 ± 7.58
**Height**	
Mean ± SD	175.40 ± 8.78
**Weight**	
Mean ± SD	75.89 ± 13.24
**BMI, kg/m^2^**	
Mean ± SD	24.5 ± 2.72
**Type of physical effort that work involve**	
Sedentary	501 (41)
Many hours of standing and walking without weight transfer	404 (33)
Many hours of standing and walking as well as lifting and transferring weights	287 (23.4)
Lifting heavy objects	32 (2.6)

**Table 2 ijerph-20-02538-t002:** Sports Training History Prior to Practicing CrossFit.

Any Sports Participation before Starting CrossFit?	
No	155 (12.7)
Yes	1069 (87)
**Type of sport practiced ***	
Soccer	145 (11.8)
Basketball	80 (6.5)
Volleyball	26 (2.1)
Running	64 (5.2)
Cycling	15 (1.2)
Swimming	57 (4.7)
Weight training	430 (35.1)
Bodybuilding	5 (0.4)
Tennis	14 (1.1)
Group training programs	93(7.6)
Functional training	30 (2.5)
Martial arts	97 (7.9)
Track and field sports	80 (6.5)
Other	83 (6.8)
**Level of sport practiced**	
Low	216 (17.6)
Medium	598 (48.9)
High	410 (33.5)

* Participants could type more than one sport.

**Table 3 ijerph-20-02538-t003:** CF participant characteristics.

Athlete Characteristics	n (%)
**CF Experience, n (%)**	
0–6 mo	126 (10.3)
6–12 mo	143 (11.7)
12–24 mo	361 (29.5)
≥24 mo	594 (48.5)
**Reasons for starting participating CF ***	
Fitness	835 (68)
Curiosity	321 (26)
Recommendation from a professional	263 (22)
**Practice frequency**	
1–2 times/wk	69 (6)
3–4 times/wk	592 (48)
>4 times/wk	563 (46)
Mean	4.27 ± 1.14
Median (range)	4 (1–7)
**Workout Duration, n (%)**	
<1 h	98 (8)
≥1 h	1126 (92)

* Participants could choose more than one answer.

**Table 4 ijerph-20-02538-t004:** Comparison of the profile characteristics of CF athletes with and without the presence of injuries.

	Presence of Injury			
Variable	Yes	No	Total Responses	*p*
**Sex**	**n (%)**	**n (%)**	**N**	0.001 *
Female	143 (32.3)	300 (67.7)	443	
Male	391 (50.1)	390 (49.9)	781	
**Age**				0.000 *
Mean (±SD)	31.79 ± 7.52	29.96 ± 7.53	30.76 ± 7.58	
**Height**				0.899
Mean (±SD)	175.43 ± 8.56	175.37 ± 8.95	175.40 ± 8.78	
**BMI**				0.000 *
Mean (±SD)	24.96 ± 2.71	24.14 ± 2.67	24.50 ± 2.72	
**Experienced in CrossFit training, y**				0.000 *
<1	67 (24.9)	202 (75.1)	269	
1–2	107 (29.6)	254 (70.4)	361	
>2	360 (60.6)	234 (39.4)	594	
**training time, min/wk**				0.001 *
<30 min	4 (44.4)	5 (55.5)	9	
30–60 min	234 (38.6)	372 (61.4)	606	
61–90 min	175 (39.9)	264 (60.1)	439	
91–120 min	46 (64.8)	25 (35.2)	71	
≥121 min	75 (75.7)	24 (24.2)	99	
**Weekly participation, d/wk**				0.000 *
<3	24 (34.8)	45 (65.2)	69	
3–5	419 (41.6)	589 (58.4)	1008	
>5	91 (61.9)	56 (38.1)	147	
**Participation in competitions**				0.001 *
yes	189 (68.5)	87 (31.5)	276	
no	345 (36.4)	603 (63.6)	948	
**Rest day**				0.001 *
≤3 times/wk	488 (45.2)	592 (54.8)	1080	
>3 times/wk	46 (31.9)	98 (68)	144	
**Level of athletic activity-fitness before starting CF**				0.001 *
Low	100 (46.3)	116 (53.7)	216	
Medium	210 (35.1)	388 (64.9)	598	
High	224 (54.6)	186 (45.4)	410	
**Mentioning medical history to CF coach before started training**				0.000 *
yes	406 (40.2)	603 (59.8)	1009	
no	128 (59.5)	87 (40.5)	215	
**Coach monitoring**				0.001 *
yes	478 (41.4)	675 (58.5)	1153	
no	55 (77.5)	16 (22.5)	71	
**Monitoring by a health professional on a regular basis**				0.000 *
yes	212 (54.4)	178 (45.6)	390	
no	322 (38.6)	512 (61.4)	834	
**Passive recovery**				0.001 *
yes	159 (63.6)	91 (36.4)	250	
no	375 (38.5)	599 (61.5)	974	
**Nutrition program**				0.001 *
yes	139 (54.5)	116 (45.5)	255	
no	395 (40.8)	574 (59.2)	969	
**Dietary supplement**				0.001 *
yes	326 (51.1)	312 (48.9)	638	
no	208 (35.5)	378 (64.5)	586	
**Types of dietary supplement *** *****				0.094
Vitamins	190 (54.3)	160 (45.7)	350	
Amino acids	140 (54.1)	119 (45.9)	259	
Protein	138 (40.6)	202 (59.4)	340	
Other	50 (68.5)	23 (31.5)	73	

* *p*-value < 0.05. **: Participants could choose more than one answers.

**Table 5 ijerph-20-02538-t005:** Descriptives of other variables in injured participants.

Variable	Injured Participants
**Report to coach about pain, discomfort or inability to execute the program**	
Yes (I mentioned it and stopped)	455 (85.2)
No (I did not mentioned it and continued)	75 (14)
**Injury to the same body location in the past**	
Yes	224 (41.9)
No	310 (58.1)
**Type of injury ***	
Muscle Pain	274 (51.3)
Tendon Pain	265 (49.6)
Joint Pain	142 (26.6)
Neuropathic Pain	73 (13.7)
I do not know the type of injury I had	41 (7.7)
**Abstinence from training (days)**	
1 to 7 days	111 (20.8)
7 to 14 days	58 (10.9)
14 to 28 days	55 (10.3)
28 to 60 days	40 (7.5)
More than 60 days	44 (8.2)
**Treat injury by a health care professional-rehabilitation specialist**	
Yes	419 (78.5)
No	115 (21.5)
**Type of health care professional ***	
Doctor	265 (49.6)
Physiotherapist	280 (52.4)
Other	9 (1.7)
**Treatment followed ***	
Physiotherapy	298 (55.8)
Rest	385 (72.1)
Medication	147 (27.5)
Injections	63 (11.8)
Surgery	22 (4.1)

* Participants could choose more than one answer.

**Table 6 ijerph-20-02538-t006:** Logistic regression model to predict injuries from five risk factors.

Risk Factors	B	S.E.	df	Sig.	Odds Ratio (95% CI)
Participation in competitions (yes/no)	1.085	0.18	1	0.00 *	2.959 (2.49–3.58)
Training Level prior to CF (Low/Medium/High)	−0.072	0.09	1	0.43 *	0.931 (0.88–0.98)
Rest/week (days/week)	0.253	0.095	1	0.01 *	1.288 (1.12–1.44)
Training Duration (minutes per session)	−0.156	0.052	1	0.00 *	0.856 (0.76–0.93)
Frequency of training (Days/week)	0.187	0.08	1	0.02 *	1.206 (1.14–1.27)
Constant	−1.286	0.587	1	0.03 *	

* *p*-value < 0.05.

## Data Availability

Not applicable.

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
