# Peer review of "Epidemiological Profile among Greek CrossFit Practitioners"

_ijerph, 2023, doi:10.3390/ijerph20032538_

Round 1
Reviewer 1 Report
1. Consider providing for continuous variables mean and standard deviation or median and P25-P75 according to the distribution of the data, not both.
2. Please avoid repeating data in the text that has already been mentioned in the tables.
3. Add a title (number/participants) to the X-axis in figure 1
4. In table 4, consider providing the mean (SD) or median (p25-p75 or RIC) according to the data distribution and provide the statistical test used to compare.
5. Please include the variables in the univariate analysis that showed significance and later in the multivariate analysis. Mention that a diagnosis of the RL model was made and that multicollinearity and interactions were evaluated.
6. In table 6 (multivariate RL model) include the units of the risk factors, OR (CI 95%).
7. Please shorten the conclusions section, and limit yourself to the aim's objective.
Author Response
Dear Reviewer,
We are sending you in response to the email regarding your comments about our manuscript (Manuscript ID: ijerph-2142193) entitled “Epidemiological Profile Among Greek CrossFit Practitioners”. We have studied the comments and uploaded the revised manuscript.
In the enclosed table all comments and corrections are listed. Please let me know if there are any other changes that need to be made.
Yours sincerely,
Ioannis A. Poulis, PT, MSc, PhD
Associate Professor, Faculty of Physiotherapy, School of Health Sciences, University of Thessaly, 3rd. Km Old National Road Lamia-Athens, GR-35132 Lamia, Greece
Email: [email protected]

Reviewer 2 Report
Abstract
I believe the abstract might include more concrete information (e.g., N injuries, N injuries per type…etc) in the results section to be more communicative. I suggest reorganizing the abstract accordingly.
Introduction
As a whole, the introduction is clear and provides quite a good rationale for the study. What I suggest is, instead of or in addition to describing the “history” and the “philosophy” of Crossfit, providing a more insight description of what crossfit actually is, what type of exercises is made of, and why the authors believe that this could lead to some injuries.
Results
- Please create a clearer figure
- I also suggest showing the most important results as figure(s) and not as table.
- The odds ratio should include confidence interval
Discussion
The discussion is overall good. I would like to see, as well within the results too, a more profound explanation about the independent factors like training level…etc. I believe this is an important issue to be fixed.
Author Response
Dear Reviewer,
We are sending you in response to the email regarding your comments about our manuscript (Manuscript ID: ijerph-2142193) entitled “Epidemiological Profile Among Greek CrossFit Practitioners”. We have studied the comments and uploaded the revised manuscript.
In the enclosed table all comments and corrections are listed. Please let us know if there are any other changes that need to be made.
Yours sincerely,
Ioannis A. Poulis, PT, MSc, PhD
Associate Professor, Faculty of Physiotherapy, School of Health Sciences, University of Thessaly, 3rd. Km Old National Road Lamia-Athens, GR-35132 Lamia, Greece
Email: [email protected]
